# Exploring the Potential of the Cost-Efficient TAHMO Observation Data for Hydro-Meteorological Applications in Sub-Saharan Africa

**Julia Schunke** [1], **Patrick Laux** [1,2,*], **Jan Bliefernicht** [1], **Moussa Waongo** [3], **Windmanagda Sawadogo** [1] and **Harald Kunstmann** [1,2]

1   Institute of Geography, University of Augsburg, 86159 Augsburg, Germany; schunke.julia@hotmail.de (J.S.); jan.bliefernicht@geo.uni-augsburg.de (J.B.); sawadogowind@gmail.com (W.S.); harald.kunstmann@kit.edu (H.K.)
2   Karlsruhe Institute of Technology (KIT), Campus Alpin, 82467 Garmisch-Partenkirchen, Germany
3   AGRHYMET Regional Centre, Training and Research Department, Niamey 11011, Niger; m.waongo@agrhymet.ne
*   Correspondence: patrick.laux@kit.edu

**Abstract:** The Trans-African Hydro-Meteorological Observatory (TAHMO) is a promising initiative aiming to install 20,000 stations in sub-Saharan Africa counteracting the decreasing trend of available measuring stations. To achieve this goal, it is particularly important that the installed weather stations are cost-efficient, appropriate for African conditions, and reliably measure the most important variables for hydro-meteorological applications. Since there exist no performance studies of TAHMO stations while operating in Africa, it is necessary to investigate their performance under different climate conditions. This study provides a first analysis of the performance of 10 selected TAHMO stations across Burkina Faso (BF). More specifically, the analysis consists of missing value statistics, plausibility tests of temperature (minimum, maximum) and precipitation, spatial dependencies (correlograms) by comparison with daily observations from synoptical stations of the BF meteorological service as well as cross-comparison between the TAHMO stations. Based on the results of this study for BF for the period from May 2017 to December 2020, it is concluded that TAHMO potentially offers a reliable and cost-efficient solution for applications in hydro-meteorology. The usage of wind speed measurements cannot be recommended without reservation, at least not without bias correcting of the data. The limited measurement period of TAHMO still prevents its usability in climate (impact) research. It is also stressed that TAHMO cannot replace existing observation networks operated by the local meteorological services, but it can be a complement and has great potential for detailed spatial analyses. Since restricted to BF in this analysis, more evaluation studies of TAHMO are needed considering different environmental and climate conditions across SSA.

**Keywords:** cost-efficient weather stations; validation; geostatistical analysis; correlogram; Trans-African Hydro Meteorological Observatory (TAHMO); sub-Saharan Africa; Burkina Faso





## 1. Introduction

Monitoring weather and climate trends is of utmost importance to provide reliable weather forecasts and climate predictions. This knowledge is necessary for agricultural planning and water management worldwide.

Despite the importance, the number of weather stations operated by national weather services has in general decreased since 1990. Some regions in Africa and South America are particularly affected, which already have a dwindling number of stations compared to industrialized countries [1–3].

When considering the weather and climate network in Africa, most of the stations are located in northern and southern Africa. Large data gaps are mainly found in central and western sub-Saharan Africa (SSA). Meteorological observations are rare and often

not available in a digital form. Thereby, several factors contribute to challenges of data accessibility, including legal restrictions, low financial investment, lack of dissemination capacity and tools, and high access costs [4].

Partly due to the lack of high quality and long term observational data in West Africa [5,6], the main processes between land-atmosphere interactions or the climatic drivers of the West African monsoon (WAM) are still not fully understood [7], making it, for instance, difficult to determine certain weather patterns resulting in extreme events. As a consequence, weather and climate forecasts are afflicted with large uncertainties, whereas the vulnerability of the population in West African is increasing due to climate change [5]. This affects food production, in general, and in particular the reliable estimation of the onset of the rainy season to support farmers management options such as the sowing date (e.g., [8–10]). An interruption of the monsoon or a wrong predicted onset of the rainy season may lead to yield reductions. These effects can become more frequent as a consequence of climate change [11,12]. The population in BF is particularly vulnerable as approximately 80% is involved in the agricultural sector, which is dominated by subsistence farming. This further underlines the importance of reliable weather and climate information [11,12].

Therefore, a densification of the hydro-meteorological measurement network is essential. In light of this need, the Trans-African Hydro-Meteorological Observatory (TAHMO) has set itself the goal to install 20,000 hydro-meteorological stations in SSA, with a maximum distance of 30 km. To achieve this goal, TAHMO cooperates with local schools and universities across SSA. The basic rationale is that a school can run and maintain a climate station and can include environmental and technical lessons into the curriculum. Thus, the younger population gets involved into the scientific work and becomes aware of environmental changes at the same time [13]. To ensure appropriateness of TAHMO stations under conditions prevailing in West African, the stations need to be reliable and cost-effective. The latter is particularly related to costs for the maintenance, which is accomplished by integrating schools and universities in the surrounding of the stations [13].

The usage of TAHMO data in climate (impact) research is still restricted due to the limited length of the measured time series. The first station was installed in 2012 [13]. Since then hundreds of stations have been installed in SSA. Thus, TAHMO might be a suitable reference data set for climate (impact) research in the near future, but it might already now be suitable for hydro-meteorological applications in SSA. Prior to its application, the quality of the measured variables needs to be ensured by comparison with the best available reference data, i.e., from stations operated by the national meteorological services, referred to as MET stations in the following.

Only few studies exist, in which TAHMO has been applied for validation of satellite product such as the Climate Hazards Group Infrared Precipitation with Stations (CHIRPS), the Tropical Applications of Meteorology Using Satellite and Ground-based Observations (TAMSAT) as well as the Integrated Multi-Satellite Retrievals for GPM early run (IMERG-ER) of the Global Precipitation Measurement Mission (GPM) (e.g., [14,15]).

However, there is no study to date that examines the performance of the TAHMO observations itself, for sites in Africa. Two studies exist, conducting performance analyses of the sensors used in the TAHMO stations (from the all-in-one weather station ATMOS 41), using in-situ reference observations have been carried out under European conditions, both of them restricted to short measurement campaigns.

Based on the study of Anand and Molnar [16], conducted at the Eidgenössische Technische Hochschule (ETH) Zurich in Switzerland using a weather station of the Institute of Atmospheric and Climate Science as a reference station, it is found that temperature biases are generally low, stemming mainly from underestimations of the cold night temperatures. Precipitation was underestimated by less than 10%, which could be mostly related to snowfall, i.e., snow is not melted by the ATMOS41 and likely partially blown out of the gauge. For wind speed, an overestimation in the order of 25% with unexplained cause is reported for ATMOS 41. The bias of relative humidity was in the order of 5%, mostly related to saturation condition, which were much more frequently observed than for the reference station.

The measured solar radiation was found to be reasonable (high correlation and a dry bias < 10%). Anand and Molnar [16] stressed the need of doing a performance analysis under environmental and climate conditions prevailing in Africa. Particular focus should be laid on high precipitation intensities occurring in tropical regions. Dombrowski et al. [17] report about relatively large biases for precipitation and wind speed. Further work should focus on the performance assessment of the ATMOS 41 during extreme precipitation and wind speed as well as the long-term durability and accuracy of the station.

In this study, the availability, plausibility, and performance of TAHMO stations is analysed for BF, as BF covers different climatic regions in West Africa. Another reason for selecting BF is that it has a relatively dense TAHMO network with a low number of missing values (MVs). Moreover, the TAHMO stations are in close distance to 10 synoptic stations of the meteorological service (MET) of BF, which can thus serve as reference stations to facilitate a performance analysis.

The objectives of this study are as follows:

- Presentation of a basic overview of the availability of TAHMO stations across West Africa, and specifically the availability of TAHMO measurements for stations in BF, since data availability is of utmost importance for this challenging region.
- Perform a cross-comparison of the TAHMO measurements using geostatistical approaches like spatial correlograms to assess, whether the spatial dependence structure of meteorological variables can be reliably reproduced by this network.
- Conduct an inter-comparison between TAHMO- and the MET stations for the variables temperature (minimum, maximum), precipitation, relative humidity, and wind speed, aiming to evaluate the reliability and quality of the TAHMO measurements.

## 2. Materials and Methods

### 2.1. Study Area

Burkina Faso (BF) is a landlocked country bordered by five countries, namely Ghana, Mali, Niger, Ivory Coast, Togo, and Benin. The climate in BF is strongly influenced by the WAM and seasons can be divided into a rainy and dry period. The dry season, from approximately November to April, is dominated by a dry and dusty wind regime (Harmattan) from the Sahara. The rainy season, in northern hemispheric summer, results from the northward shift of the Intertropical Convergence Zone (ITCZ) and transition of the wind regime with its maximum in August [18]. The characteristics and intensity of the monsoon is a very complex interplay of global atmospheric and oceanic circulation processes and is still in focus of research. BF covers three climate zones, from the Sahel in the north through the Sudan-Sahel to the Sudan in the south, which are characterised by rainy seasons of varying lengths [18,19].

### 2.2. Data Availability

#### 2.2.1. TAHMO Data

TAHMO operates the all-in-one weather station ATMOS 41, developed by the METER Group, consisting of sensors for measuring solar radiation, precipitation, relative humidity, air temperature, vapour pressure, barometric pressure, wind gust, wind speed, wind direction, and lightning. Further characteristics about ATMOS41 and the sensors are summarized in van de Giesen et al. [13], Anand and Molnar [16], Dombrowski et al. [17] and the technical manual [20]. Note that due to the lack of reliable reference data for solar radiation, vapour pressure, barometric pressure, wind gust, wind direction, and lightning, the evaluation of these variables is not considered in this study.

Information about the retrieval of the data can be obtained from https://tahmo.org/ (last access: 22 November 2021). To ensure appropriateness of TAHMO stations under conditions prevailing in West African, the stations need to be reliable and cost-effective. The latter is particularly related to costs for the maintenance, which is accomplished by integrating schools and universities in the surrounding of the stations [13].

In the following the data availability of the TAHMO network for West Africa is elaborated. This covers spatial and temporal aspects such as the spatial distribution of the stations and the number as well as the temporal distribution of missing values. A map of the spatial distribution of TAHMO stations across SSA can be found in Figures 1 and A1.

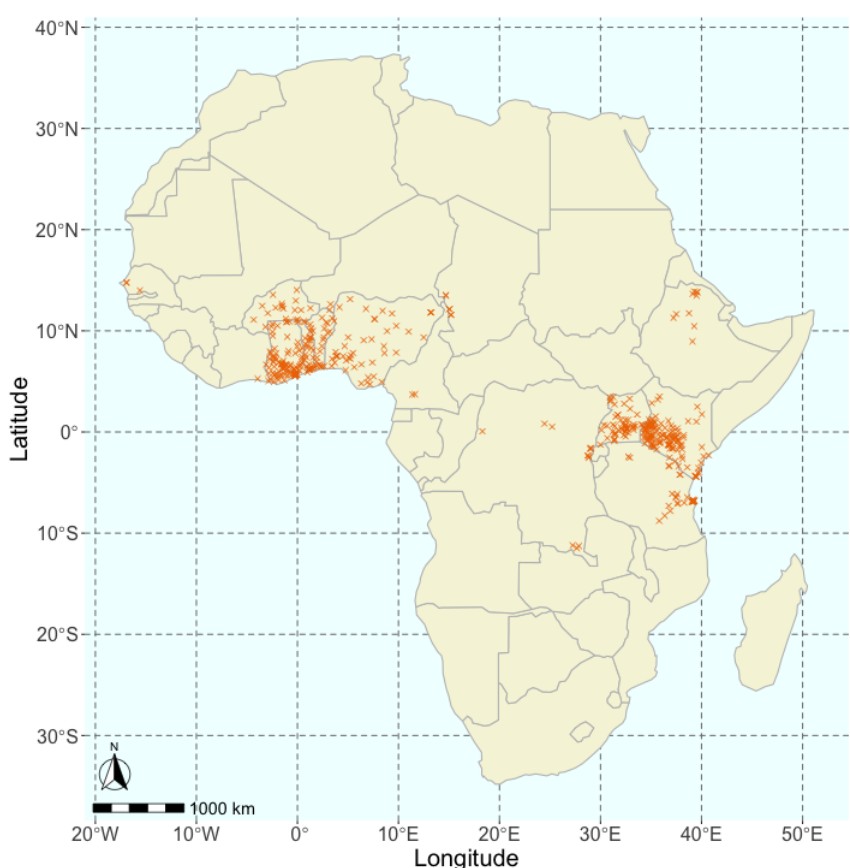

**Figure 1.** Location of the available TAHMO stations in sub-Saharan Africa (SSA).

In total, 219 TAHMO stations are located in West Africa within the 8 countries: BF, Benin, Republic Côte d'Ivoire, Ghana, Niger, Nigeria, Senegal and Togo. Most of the stations are located in Ghana (99 stations), followed by Nigeria (45 stations), and Togo (30 stations). There are 17 TAHMO stations in BF.

In the focal region (i.e., BF) of this study, the TAHMO network consists of 17 irregularly distributed stations that covers different climate zones (see Figure 2). First recordings in BF are available from May 2017. Out of the 17 available TAHMO stations for BF, stations with a measurement period of less than 2 years have been sorted out. An overview of the start of the measurements at each station in BF is shown in Figure A2. The remaining 10 stations are in close distance to the MET stations and are available from May 2017.

Concerning the missing value statistics, Figure 3 illustrates the summarized percentage data availability of all variables from the 10 selected stations in BF within the period 05/21/2017–12/31/2020. In general, it can be stated that wind speed, wind direction and wind gust have the highest number of MVs with MVs being higher than 5% for all stations. The lowest number of MVs can be found for temperature, solar radiation, and relative humidity (with less than 15% MVs in general, and MVs with less than 1% for 5 stations). The number of MVs differs significantly for the stations (with TA00167, TA00168, TA00163 having no MVs for all variables except wind-related measurements. In turn, Pô (TA00165) has the highest number (approximately 13%) of MVs for most variables, wind-related variables have >20% of MVs.)

### 2.2.2. Reference Data

10 synoptical stations of the Agence Nationale de la Météorologie (ANAM) of Burkina Faso serve as reference. These stations are distributed relatively homogeneously across the country (Figure 2). In addition to the synoptical stations, ANAN operates other meteorological networks (e.g., the rainfall network and agrometeorological network). However, the data from the synoptical stations are of highest quality, contain less data gaps, and many different meteorological variables are measured jointly at the same locations following the World Meteorological Organization (WMO) standards. The synoptical data was recently provided by ANAM for the variables precipitation, temperature (minimum, maximum), relative humidity (minimum, maximum, mean), wind speed, mean sea level pressure, and sunshine duration. The data is in daily resolution and is provided for a period of either 30 years (1.1.1991 to 31.12.2020, for the stations Ouagadougou, Dori, Dédougou and Pô) or 10 years (1.1.2011 to 31.12.2020, for the stations Ouahigouya, Bogande, Fada N'Gourma, Bobo-Dioulasso, Boromo and Gaoua). In this study, we use daily precipitation amount, daily maximum and minimum temperature at 2 m, daily mean relative humidity at 2 m, and daily mean wind speed at 10 m for comparison with the TAHMO measurements. Minimum, maximum and mean values refer to a time window from 00:00 UTC (previous day) to 00:00 UTC for a given measurement day.

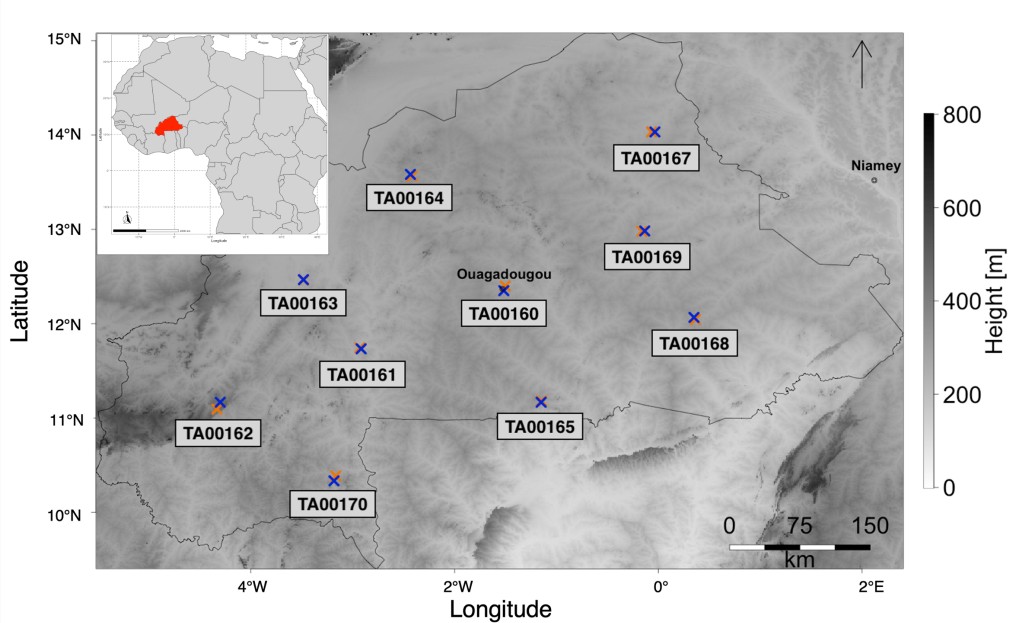

**Figure 2.** Topographic map from Burkina Faso with the examined stations and the TAHMO station name (for MET station name see Table 1). Stations from the Burkina Faso meteorological service are in blue, the TAHMO stations are highlighted in orange. The TAHMO stations with the largest distance to the reference MET station are in Ouagadougou, Bobo-Dioulasso and Gaoua. Exact distance values can be found in Table 1.

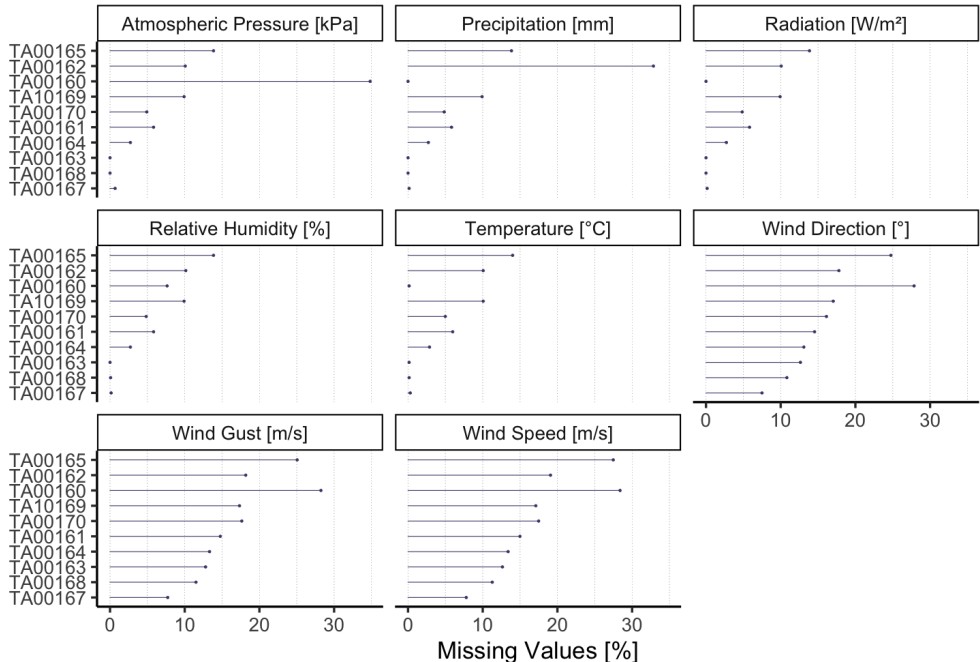

**Figure 3.** Number of missing values (MVs) (%) for all variables measured by TAHMO in Burkina Faso within the period 06/20/2017–12/31/2020, and the pre-selected 10 hydro-meteorological TAHMO stations in Burkina Faso.

The MV statistics of the MET stations is given in Figure A3. In general, the number of MVs is not critical for comparison. Minor restrictions are found for wind speed, where the number of MVs is exceeding 30% for station Bogandé, but <15% for all other stations.

**Table 1.** TAHMO station code and name, the corresponding meteorological station name and the distance between those two stations is listed. These 10 TAHMO stations are compared to 10 meteorological reference stations (MET stations).

| TAHMO Station Code and Name | MET Station Name | Distance (m) |
| --- | --- | --- |
| TA00160 Station Somgande Meteo | Ouagadougou aéro | 5700 |
| TA00161 Boromo | Boromo | 1800 |
| TA00162 Farakobo | Bobo-Dioulasso | 8750 |
| TA00163 Dédougou | Dédougou | 380 |
| TA00164 Ouahigouya | Ouahigouya | 2680 |
| TA00165 Pô | Pô | 1580 |
| TA00167 Dori | Dori | 3520 |
| TA00168 Fada | Fada Ngourma | 2850 |
| TA10069 Bogandé | Bogandé | 3020 |
| TA00170 Gaoua | Gaoua | 6100 |

### 2.3. Plausibility Checks

In order to perform plausibility tests, daily precipitation and minimum temperature time series of all TAHMO stations in BF were plotted as time series (see Figure 4). The following criterions are considered for this region:

- Temperature should not fall below 0 °C
- Dry and rainy seasons should be observable in the rainfall patterns
- Annual precipitation should decrease towards the north, while remaining close to the climatological amounts
- Daily rainfall amounts >500 mm are not realistic

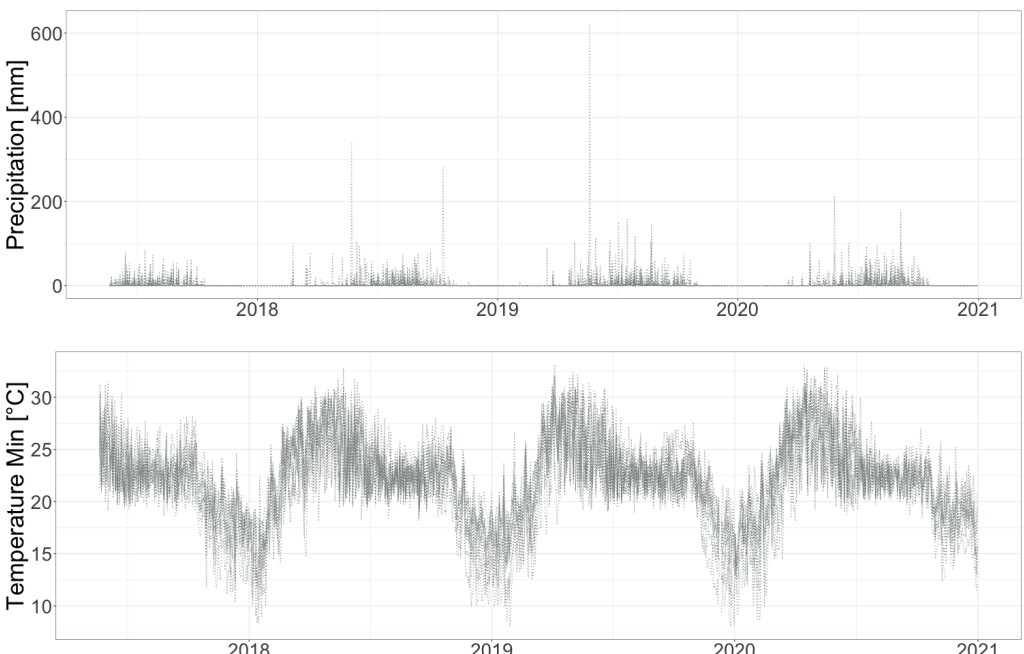

**Figure 4.** Daily precipitation and minimum temperature time series of all TAHMO stations in Burkina Faso to perform plausibility checks.

If all these assumptions are fulfilled, the station can be seen as plausible on first sight. Since no station has below 0 °C (with lowest temperature values around 9 °C), the first assumption can be confirmed. The rain and dry seasons are plausibly measured between May to September, depending on the latitude of the station location. The longest rainy season appears to be in southernmost station TA00170. At station TA00164, outliers become visible with one value being higher than 600 mm. Additionally, the 2018 and 2019 precipitation values are accumulated to annual values and compared Table 2). Only stations with no MVs in one year were considered, for this reason TA00162 is excluded from this comparison. Station TA00164 is also excluded due to the higher number of outliers within precipitation time series. The results indicate that the amount of precipitation decreases towards the north with approx. 470 mm at the northernmost station in Dori (TA00167) and approx. 1230 mm in Gaoua (TA00170). Therefore plausibility in terms of precipitation is mostly given.

**Table 2.** Annual cumulative precipitation sum (mm) from 2018 and 2019. The amount of precipitation decreases towards the north. "–" shows that missing values (MVs) are present and can therefore not be included in the comparison. Because of the high number MVs in both years, TA00162 was completely neglected. Station TA00164 is also not involved due to the higher number of outliers within precipitation time series.

| TAHMO | 2018 (mm) | 2019 (mm) | MET Station | 2018 (mm) | 2019 (mm) |
|-------|-----------|-----------|-------------|-----------|-----------|
| TA00170 | – | 1232 | Gaoua | 1414 | 1412 |
| TA00165 | – | 1467 | Pô | 1052 | 1043 |
| TA00161 | 1016 | 1149 | Boromo | 1024 | 1088 |
| TA00163 | 841 | 1292 | Dédougou | 835 | 969 |
| TA00160 | 952 | 826 | Ouagadougou | 860 | 853 |
| TA00168 | 618 | 774 | Fada N'Gourma | 693 | 711 |
| TA10069 | 452 | 541 | Bogandé | 562 | 543 |
| TA00167 | 400 | 467 | Dori | 494 | 571 |

*2.4. Statistical Analyses*

We use several statistical approaches and measures for qualitative and quantitative analysis of the TAHMO measurements. The comparison of the TAHMO measurements with the reference MET station data is done based on visual analysis such as time series diagrams and scatter plots. In addition, several standard measures are calculated like the Pearson correlation coefficient (r), the Mean Absolute Error (MAE) and the Root Mean Square Error (RMSE) to determine the correspondence between the TAHMO and the MET station measurements in a quantitative way. r measures the strength and direction of the linear relationship between the two data sets, and MAE is the absolute measure of the differences and is thus neither subject to cancellation errors nor is it particularly sensitive to outliers. In turn, the RMSE gives more weight to larger differences between the two data sets. Further information regarding these measures is given in Wilks [21] and Jolliffe [22]. The Spearman correlation coefficient also serves as basis for the geostatistical analysis applied in this study. Here, a pairwise comparison of the TAHMO time series is done for each meteorological variable using the correlation measure. In addition, the Euclidean distance is calculated for each station pair. The r values are then related to the corresponding distances and these data points are visualized as a scatter diagram. The resulting pattern is used for computing spatial correlograms to estimate the spatial dependence structure of meteorological variables [23–25] and can provide additional information about data reliability independently from the reference data set. It is expected that the correlation decreases with increasing distance (Tobler's first law of geography, Tobler [26]), and the correlation is expected to decrease faster for variables that are spatially more variable, such as precipitation or wind speed, compared to more stable variables, such as temperature.

## 3. Results and Discussion

After the data are preprocessed and checked for plausibility, the data are analyzed further. The analyses include an inter-comparison between the TAHMO stations and a comparison of the TAHMO stations with reference stations.

*3.1. Inter-Comparison of TAHMO Stations*

In a first step, the 10 preselected TAHMO stations are compared among themselves. Figure 5 shows the Pearson correlation coefficient between each TAHMO station as a function of the distance. In general, it can be stated that nearby stations have a higher correlation than distant ones. Compared to maximum & minimum temperature as well as relative humidity, the correlation is, in general, reduced for precipitation. For wind speed, no clear spatial dependency can be observed. This is due to the high spatial variability of both, precipitation and wind speed, resulting in no or less pronounced patterns or spatial dependency. Wind speed is additionally afflicted with higher measurement uncertainties accompanied by the higher number of outliers and MVs. Considering all variables, the general tendency is visible that more distant stations show a lower correlation.

In the following the results of the comparison with reference data are demonstrated.

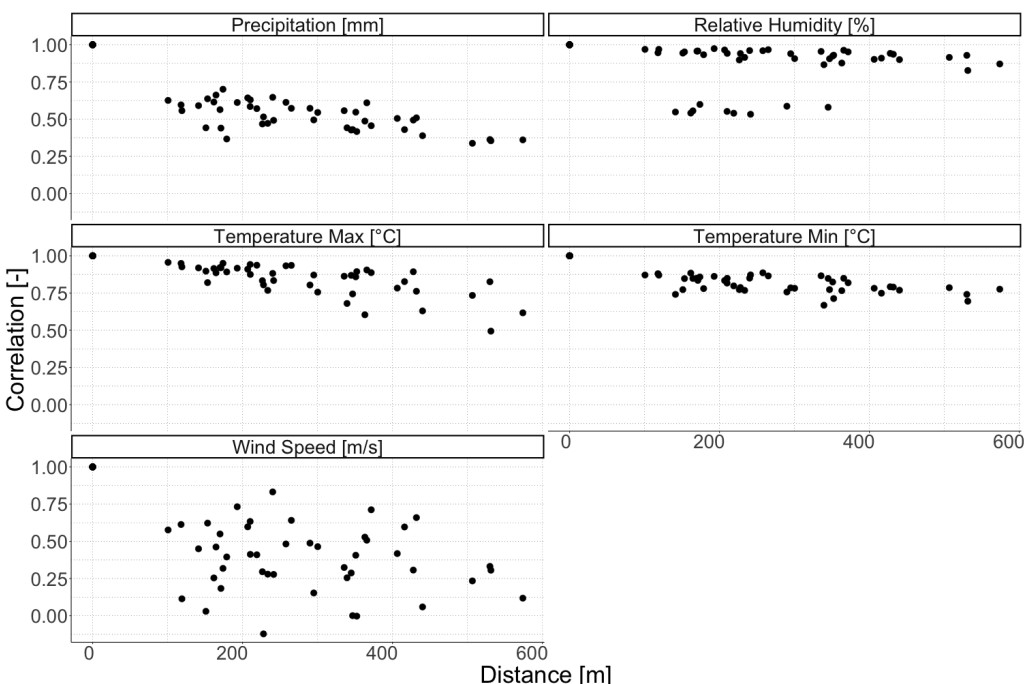

**Figure 5.** Distance-correlation relationship for examined TAHMO station in Burkina Faso and the variables maximum temperature (°C), minimum temperature (°C), precipitation (mm), relative humidity (%) and wind speed (m/s). A decrease in correlation with increasing distance becomes apparent within each variable.

### 3.2. Comparison with Stations from the BF Meteorological Service

In the following, a performance analysis of the 10 TAHMO stations for the variables maximum temperature, minimum temperature, precipitation, relative humidity, and wind speed is conducted by comparing them with their corresponding (i.e., the nearest, see Figure 2 and Table 1) reference station from meteorological service of Burkina Faso. This is done based on different performance measures for the 10 TAHMO stations. As a first step, in order to check the impact of the seasonality on the results we calculated the correlation coefficients exemplary for station Dédougou (TA00163) for dry and wet seasons, respectively (see Table 3). It is found that for precipitation the correlation during the rainy season is higher than for the dry season (0.71, 0.61, 0.73 for the rainy season, dry season, without considering seasonality, respectively). For minimum temperature, the correlation coefficient is higher for the dry season compared to the rainy season (0.83, 0.99, 0.95 for the rainy season, dry season, no discrimination, respectively). For other variables, the impact is rather negligible. Similar impacts are found for the other stations (not shown). Table 4 provides the correlation, the RMSE and the MAE for each station without considering seasonality.

**Table 3.** Correlation coefficients for Station Dédougou (TA00163) for dry and wet seasons (wet season: 21/05–20/10, dry season: 21/10–20/05) compared to the complete time series.

| Variable | Rainy Season | Dry Season | Complete Time Series |
|---|---|---|---|
| Precipitation | 0.71 | 0.61 | 0.73 |
| Relative Humidity | 0.93 | 0.94 | 0.98 |
| Temperature Max | 0.98 | 0.99 | 0.99 |
| Temperature Min | 0.83 | 0.99 | 0.95 |
| Wind Speed | 0.7 | 0.71 | 0.74 |

**Table 4.** Statistical performance measures of all variables between TAHMO and Meteorological Service of Burkina Faso. * The correlation coefficients are tested for statistical significance at $\alpha = 0.05$, and the null hypothesis (correlation coefficient equals zero) was rejected for all computed values, indicating their statistical significance.

| Station | RMSE | MAE | Correlation * |
|---|---|---|---|
| Temperature Max (Min) | | | |
| Ouagadougou | 0.93 (2.75) | 0.67 (2.18) | 0.97 (0.92) |
| Boromo | 0.43 (0.92) | 0.29 (0.45) | 0.99 (0.96) |
| Bobo-Dioulasso | 0.79 (4.36) | 0.66 (3.18) | 0.97 (0.71) |
| Dédougou | 0.87 (0.97) | 0.72 (0.45) | 0.99 (0.95) |
| Ouahigouya | 0.87 (1.13) | 0.80 (0.51) | 0.99 (0.95) |
| Pô | 0.72 (0.91) | 0.62 (0.49) | 0.99 (0.95) |
| Dori | 1.41 (1.34) | 1.28 (0.79) | 0.99 (0.96) |
| Fada N'Gourma | 0.74 (1.11) | 0.65 (0.73) | 0.99 (0.96) |
| Bogandé | 0.87 (1.12) | 0.80 (0.53) | 0.99 (0.96) |
| Gaoua | 1.28 (1.17) | 0.61 (0.69) | 0.93 (0.94) |
| Precipitation | | | |
| Ouagadougou | 9.09 | 2.44 | 0.66 |
| Boromo | 5.95 | 1.47 | 0.74 |
| Bobo-Dioulasso | 10.76 | 2.84 | 0.47 |
| Dédougou | 7.39 | 1.84 | 0.73 |
| Ouahigouya | 23.62 | 3.43 | 0.69 |
| Pô | 9.37 | 2.47 | 0.74 |
| Dori | 6.06 | 1.46 | 0.68 |
| Fada N'Gourma | 9.27 | 1.78 | 0.71 |
| Bogandé | 4.48 | 0.91 | 0.76 |
| Gaoua | 7.60 | 2.12 | 0.71 |
| Relative Humidity | | | |
| Ouagadougou | 28.67 | 19.09 | 0.62 |
| Boromo | 5.27 | 4.29 | 0.99 |
| Bobo-Dioulasso | 13.68 | 11.52 | 0.94 |
| Dédougou | 5.72 | 4.81 | 0.98 |
| Ouahigouya | 4.18 | 3.39 | 0.99 |
| Pô | 7.19 | 5.86 | 0.98 |
| Dori | 8.24 | 6.45 | 0.95 |
| Fada N'Gourma | 5.37 | 4.57 | 0.99 |
| Bogandé | 5.62 | 4.41 | 0.98 |
| Gaoua | 9.07 | 6.49 | 0.94 |
| Wind Speed | | | |
| Ouagadougou | 1.85 | 1.70 | 0.52 |
| Boromo | 0.45 | 0.34 | 0.62 |
| Bobo-Dioulasso | 2.05 | 1.77 | 0.22 |
| Dédougou | 0.91 | 0.72 | 0.74 |
| Ouahigouya | 0.66 | 0.53 | 0.69 |
| Pô | 0.83 | 0.65 | 0.19 |
| Dori | 1.16 | 1.04 | 0.66 |
| Fada N'Gourma | 1.20 | 1.07 | 0.54 |
| Bogandé | 1.03 | 0.86 | 0.80 |
| Gaoua | 0.61 | 0.46 | 0.63 |

### 3.2.1. Temperature

Figures 6 and 7 show the maximum and minimum temperature time series comparison between TAHMO stations and the corresponding stations from meteorological service of Burkina Faso. Partly large periods of MVs are visible especially for the stations Boromo, Bobo-Dioulasso, Pô, Bogandé, and to a lesser extent for Gaoua. In terms of maximum temperature, there exists an overall good agreement, but it is found that TAHMO records are constantly slightly lower in Dédougou, Ouahigouya, Pô, Dori, Fada N'Gourma and Bogandé. The total MAE for all measurements is 0.71 °C.

Similar observations are made for the time series of minimum temperature with correlation coefficients being higher than 0.94, except for Ouagadougou and Bobo-Dioulasso. Compared to the maximum temperature, the minimum temperature of TAHMO has higher deviations from the reference stations. The highest deviations can be found at Bobo-Dioulasso. This relatively higher spread of the values is also visible in Figure 8, showing the scatterplots between minimum temperature time series of TAHMO stations and the corresponding stations from meteorological service of Burkina Faso. It is obvious that the scatter is higher for the minimum temperature compared to the maximum temperature values (see Figure 9), with the highest spreads in mid-range of the minimum temperature values. The statistics given in Table 4, corroborating the above finding. Based on this, the highest inconsistencies are visible in Bobo-Dioulasso with a MAE of approximately 3.2 and a correlation coefficient of approximately 0.7.

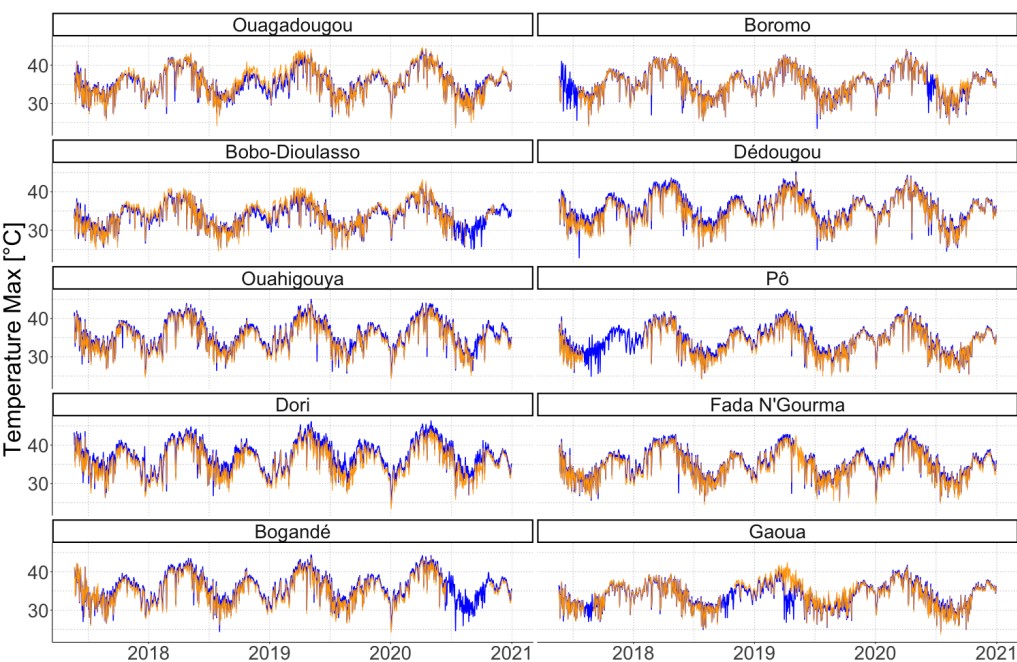

**Figure 6.** Comparison of maximum temperature time series between TAHMO stations (in orange) and the corresponding stations from meteorological service of Burkina Faso (in blue). An overall good agreement becomes apparent between the two time series. Large MVs are visible especially for the stations Boromo, Bobo-Dioulasso, Pô, Bogandé and to a lesser extend for Gaoua.

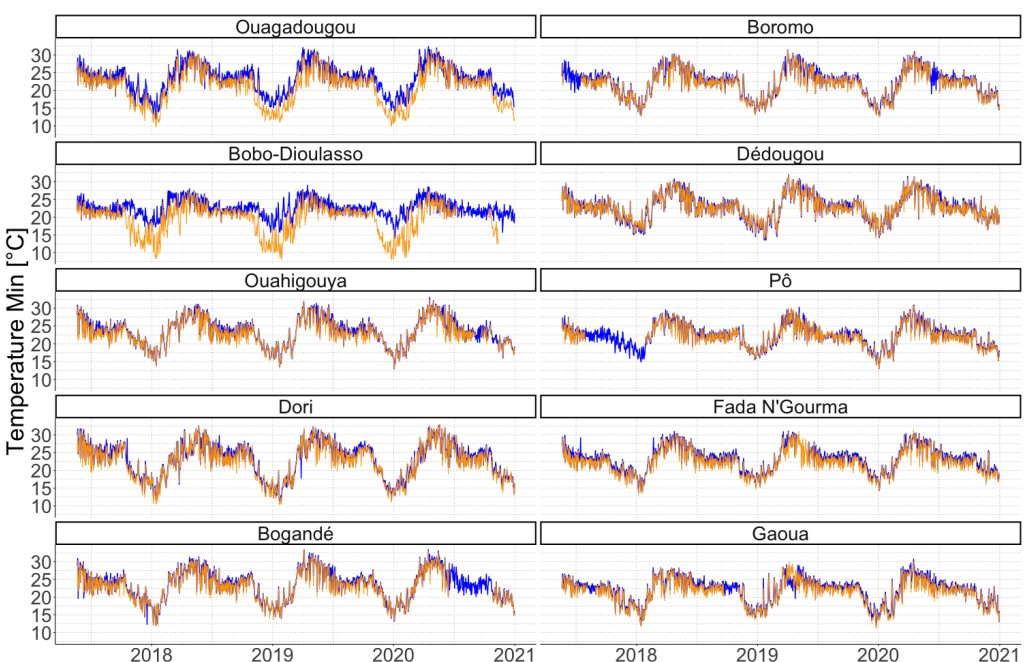

**Figure 7.** Comparison of minimum temperature time series between TAHMO stations (in orange) and the corresponding stations from meteorological service of Burkina Faso (in blue).

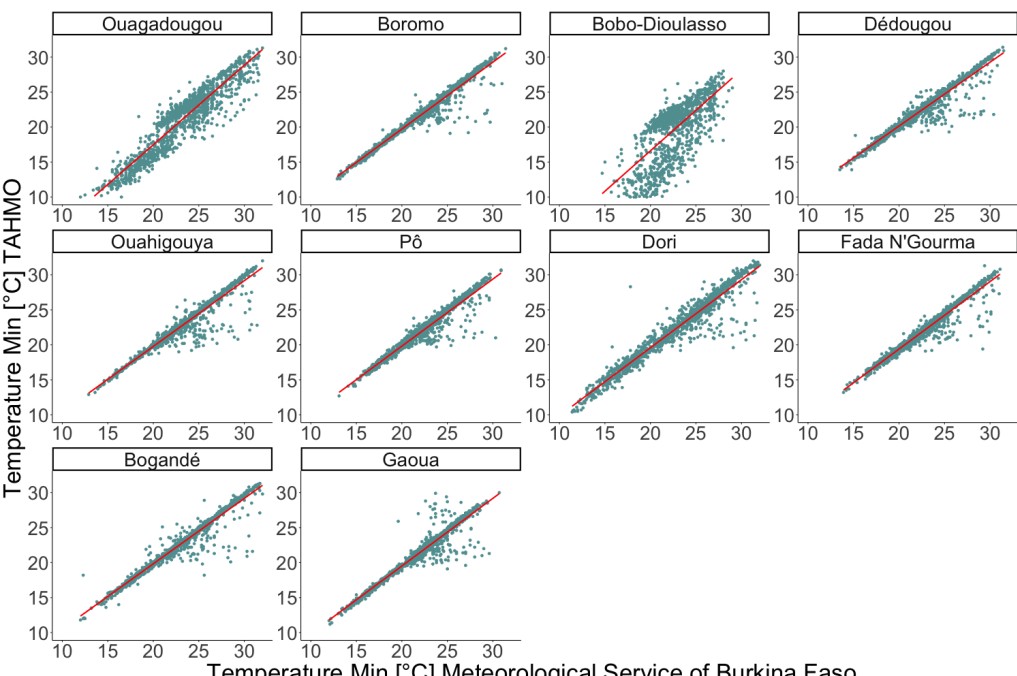

**Figure 8.** Scatterplot between minimum temperature time series of TAHMO stations (*y*-axis) and the corresponding stations from meteorological service of Burkina Faso (*x*-axis).

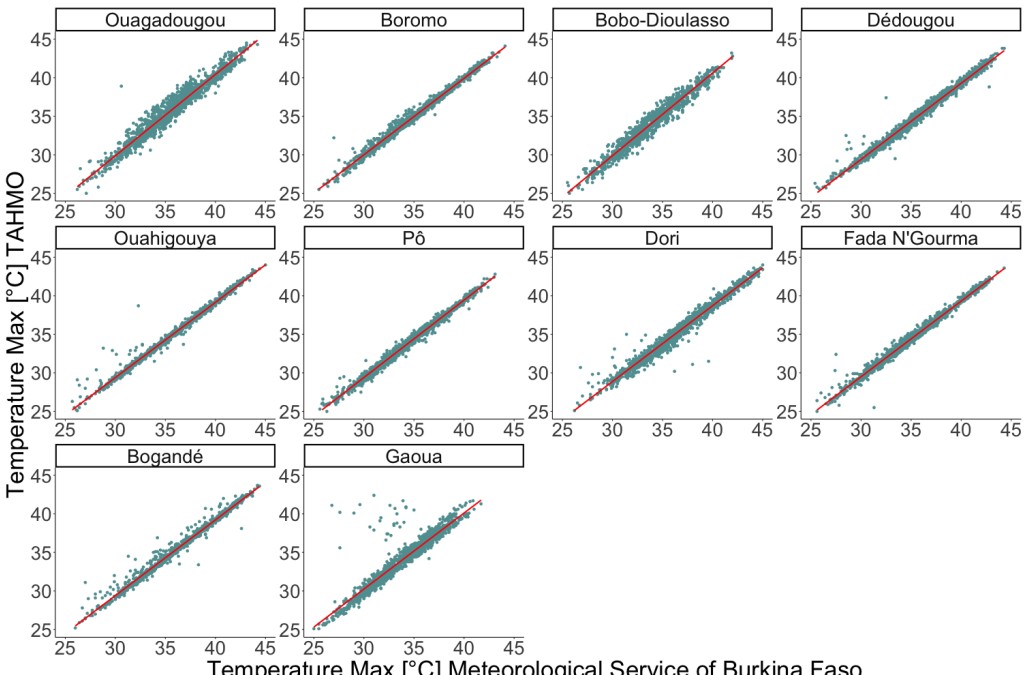

**Figure 9.** Scatterplot between maximum temperature time series of TAHMO stations (*y*-axis) and the corresponding stations from meteorological service of Burkina Faso (*x*-axis).

### 3.2.2. Precipitation

Figure 10 shows precipitation as measured by all stations compared to measurements from the meteorological service. As with the temperature measurements, large number of MVs become apparent with whole rainy seasons being not recorded (Bobo-Dioulasso in 2018 and 2020; Bogandé 2020) or incomplete (Boromo 2017; Bobo-Dioulasso 2019; Ouahigouya 2018; Pô 2017; Gaoua 2018). Most stations agree in terms of the rainy and dry seasons but differ in some cases significantly in the amount of precipitation (magnitude) and the onset of the monsoon. All correlations are in the range between 0.66 and 0.76 and agree reasonably well, with the exception of Bobo-Dioulasso with only 0.47. Bobo-Dioulasso is also the time series with the largest data gaps (Table 4). Figure 11 aims to emphasize the onset of the monsoon in 2019 highlighting different starts in each location. Stations in the southern part start earlier (Gaoua, Pô, Bobo-Dioulasso) compared to stations in northern BF (Ouahigouya, Bogandé, Dori). Regarding Ouahigouya, Ouahigouya seems to be the location with the lowest annual rainfall but the highest daily amounts. This also points to outliers with one TAHMO record being higher than 600 mm per day and could explain the high RMSE of 23.62 as RMSE is sensitive to outliers. The differences in magnitude are also visible when looking at the scatterplots presented in Figure 12. Measurements show a strong spread with an increase at higher precipitation values. The stations in Bogandé and Boromo have the most reliable results.

In order to quantify the accumulated differences between the different precipitation measurements, cumulative precipitation is calculated and displayed. This so-called double sum analysis is shown in Figure 13. The results clearly show that the gauges agree reasonably well in Ouagadougou, Dédougou, and Fada N'Gourma. The other stations reveal partly strong over- or underestimations. MVs during the rainy season heavily impact on these results, clearly visible for e.g., Bobo-Dioulasso. Measurements in Boromo, Dori, Bogandé and Gaoua differ from the beginning with TAHMO recording less precipitation from the start. Nonetheless, the stations having complete time series represent the seasonal fluctuations as good as the MET stations. Ouahigouya and Pô show possible artifacts within the TAHMO measurements. The total precipitation amounts partly vary remarkably between TAHMO and MET stations. Only three stations with no MVs, i.e., Ouagadougou, Dédougou, and Dori, have been included in this comparison. The difference in total amount

(MET station—TAHMO) are −92,330 mm (4.3%) in Ouagadougou, 100,505 mm (−4.3%) in Dédougou, and 221,799 mm (−25.4) in Dori. The differences in Ouagadougou and Dédougou are surprisingly low in regard to the long period and, especially in Ouagadougou, in regard to the distance between the TAHMO and reference station (5700 m). As the distance between the two stations in Dédougou is only 380 m and the time series contains no MVs, further more detailed analyses should be done with a lower temporal resolution. In Dori, the difference is remarkably high. As mentioned above, those two stations differ from the beginning with the TAHMO stations, which are recording less precipitation.

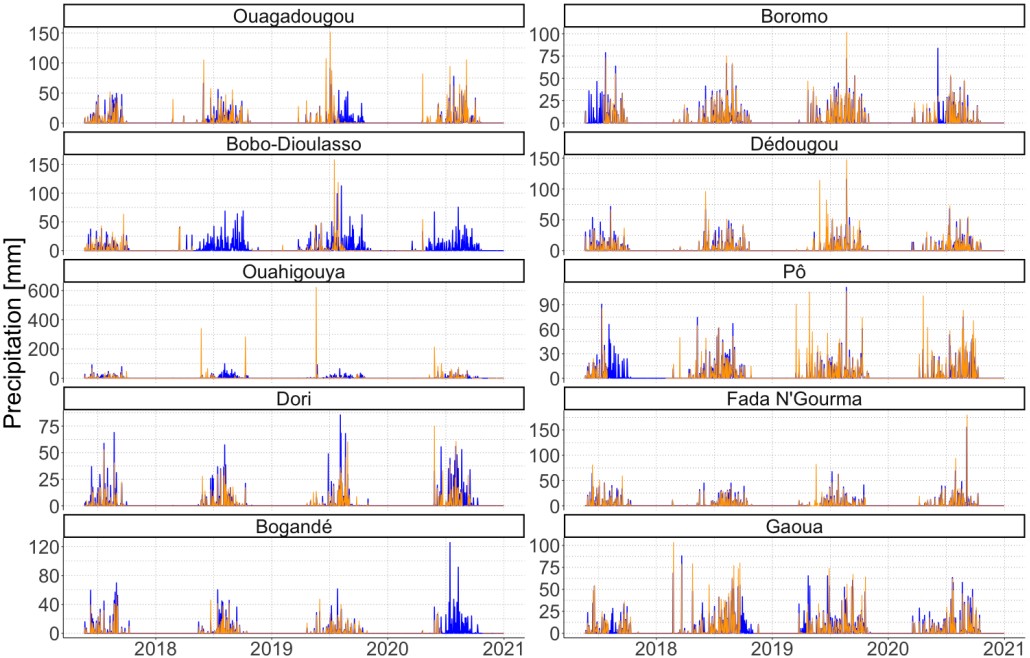

**Figure 10.** Comparison of precipitation time series between TAHMO stations in orange and the corresponding stations from meteorological service of Burkina Faso in blue. Due to large number of MVs, rainy seasons are not recorded entirely for Bobo-Dioulasso (2018 and 2020), Bogandé (2020), or only partly for Boromo (2017), Bobo-Dioulasso (2019), Ouahigouya (2018), Pô (2017), and Gaoua (2018).

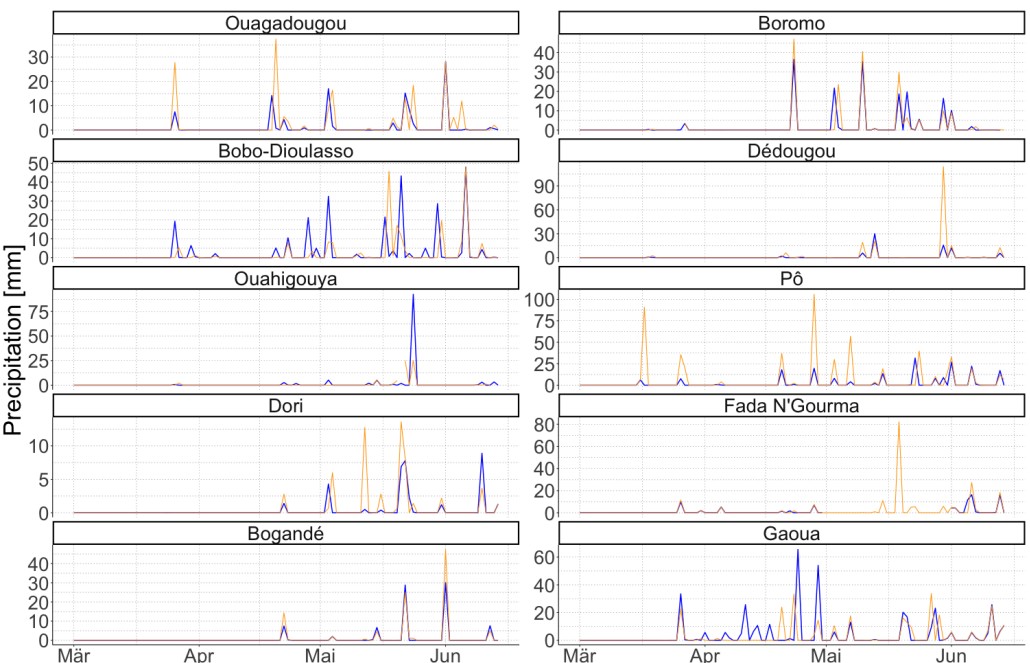

**Figure 11.** Comparison of precipitation time series between TAHMO stations in orange and the corresponding stations from meteorological service of Burkina Faso (BF) in blue from 2019-03-01–2019-06-15 to emphasize the different onsets of the monsoon in 2019. Stations in the southern part start earlier (Gaoua, Pô, Bobo-Dioulasso) compared to stations in northern BF (Ouahigouya, Bogandé, Dori).

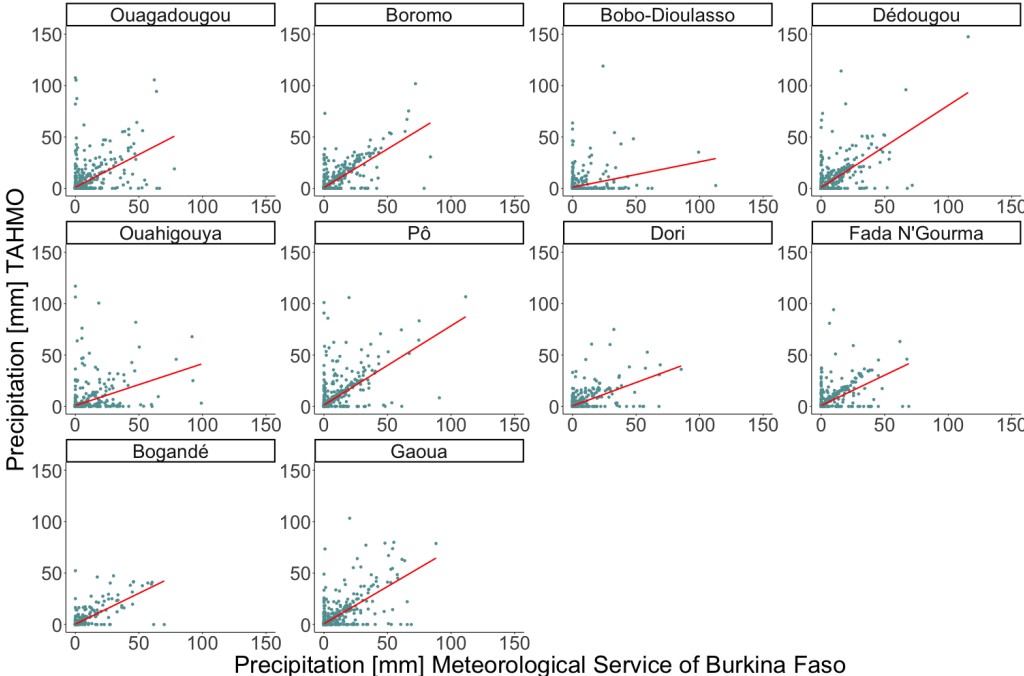

**Figure 12.** Scatterplot between precipitation time series of TAHMO stations (*y*-axis) and the corresponding stations from meteorological service of Burkina Faso (*x*-axis).

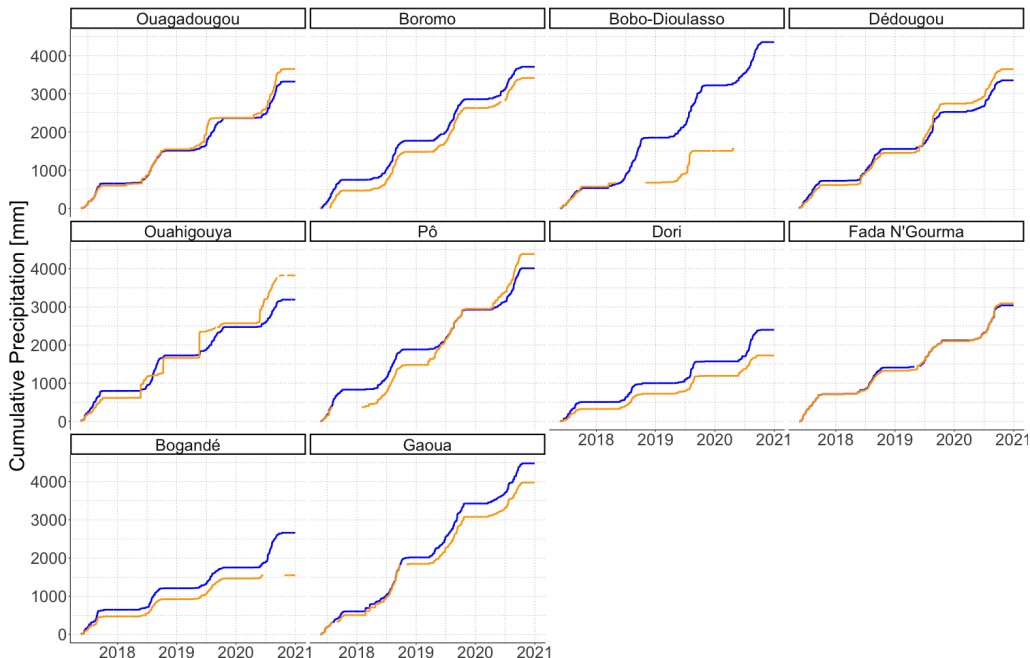

**Figure 13.** Comparison of cumulative precipitation between TAHMO stations in orange and the corresponding stations from the meteorological service of Burkina Faso in blue.

### 3.2.3. Relative Humidity

Figure 14 shows the relative humidity time series comparison between TAHMO stations and the corresponding stations from meteorological service of Burkina Faso. With exception of Ouagadougou, there is a high correlation between the two time series. Ouagadougou stands out due to large outliers in 2017 and 2018. In all the time series, the rainy and dry seasons are well defined with a higher relative humidity during the rainy seasons. It is found that the TAHMO values are slightly higher during the rainy season and slightly lower during the dry season compared to the MET station data. The same findings are also described in Dombrowski et al. [17] and Anand and Molnar [16]. This observation is in agreement with the above-mentioned systematically reduced temperature values. According to Dombrowski et al. [17], this could be a result of a wet, exposed temperature sensor or its immediate surroundings, making it more prone to evaporative cooling. The scatterplots (Figure 15) confirm this evaporative cooling effect in the TAHMO measurements. The high deviations in the scatterplot of Gaoua are due to outliers within the MET station time series. Ouagadougou also shows artificial effects in the time series: saturation at 100% is reached frequently in the TAHMO series in 2017 and 2018, but not at all within the MET station time series. These artifacts of ATMOS 41 sensors, which are used for TAHMO, are also reported in Dombrowski et al. [17] and Anand and Molnar [16].

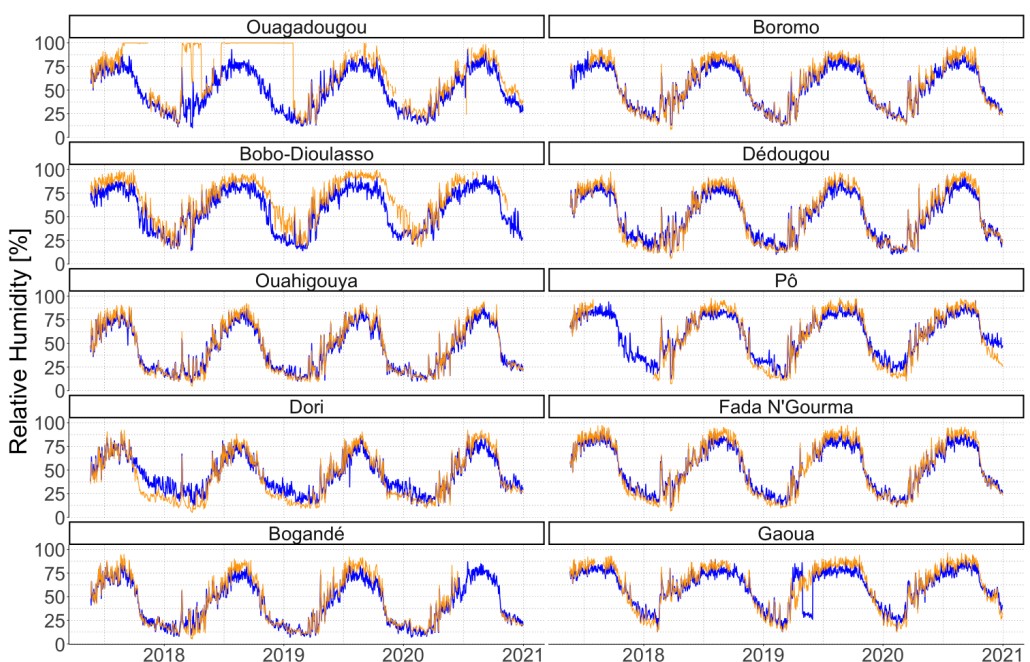

**Figure 14.** Comparison of relative humidity time series between TAHMO stations in orange and the corresponding stations from meteorological service of Burkina Faso in blue.

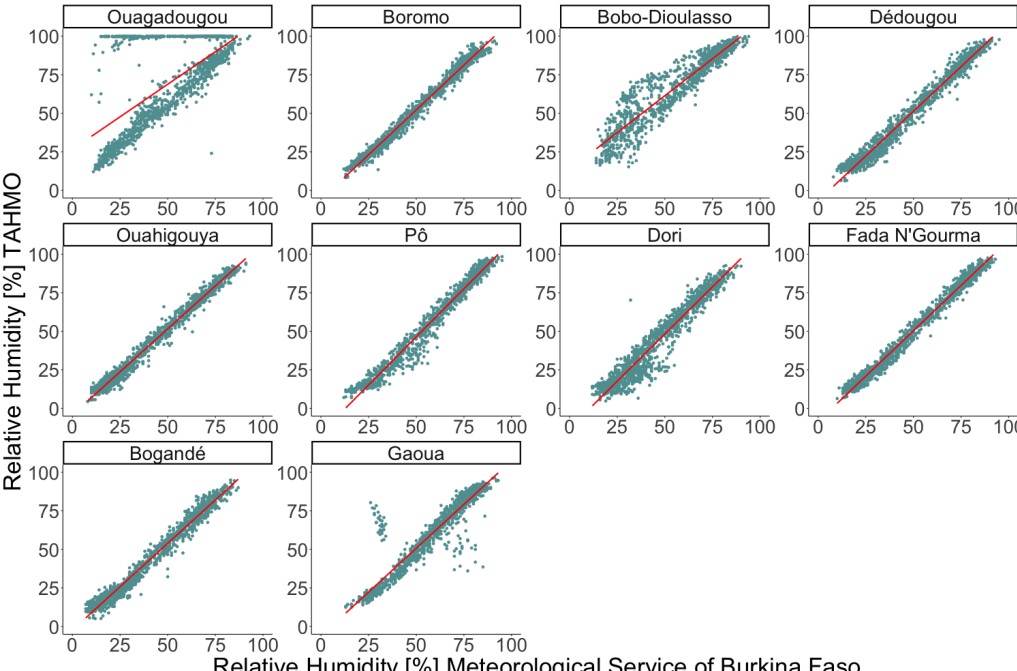

**Figure 15.** Scatterplot between relative humidity time series of TAHMO stations (*y*-axis) and the corresponding stations from meteorological service of Burkina Faso (*x*-axis). For TA00160 in Ouagadougou, a large number of artifacts can be observed.

### 3.2.4. Wind Speed

Figure 16 shows the time series of wind speed for TAHMO stations and the corresponding stations from meteorological service of Burkina Faso. The results indicate that the measurements do not agree as well as for the other variables. The variability, and particularly short-term fluctuation, is much higher in time series of the MET stations compared to TAHMO. With few exceptions only, two different patterns emerge. TAHMO measurements

are continuously lower (systematic underestimation) and have a less pronounced magnitude (lower variability). Note that the wind speed for the reference stations is measured at 10 m, whereas TAHMO measures at 2 m. Due to the higher wind velocities at higher levels (above surface), it is expected that TAHMO will measure constantly higher wind speed values, but this could (at least partly) explain the reduced short-term fluctuations compared to the reference measurements. Exceptions can be found in Dori, where the TAHMO records are on average 1.04 m/s higher than the records from the MET stations. While the TAHMO values follow the above-mentioned patterns in Bobo-Dioulasso, the TAHMO values for 2019 are systematically overestimating, explaining the high RMSE of 2.05 (see Table 4). Dombrowski et al. [17], applying the same height for the reference measurements, highlighted that both the fluctuations and the measured velocities are higher for ATMOS 41 records compared to those from the reference stations. Correlations are between 0.19 and 0.80 with the highest agreement in Bogandé (0.80) and Dédougou (0.74). Pô and Bobo-Dioulasso have by far the lowest correlation with only 0.19 and 0.22, respectively. Pô has many MVs and Bobo-Dioulasso measures continuously lower values. Scatterplots in Figure 17 support these results.

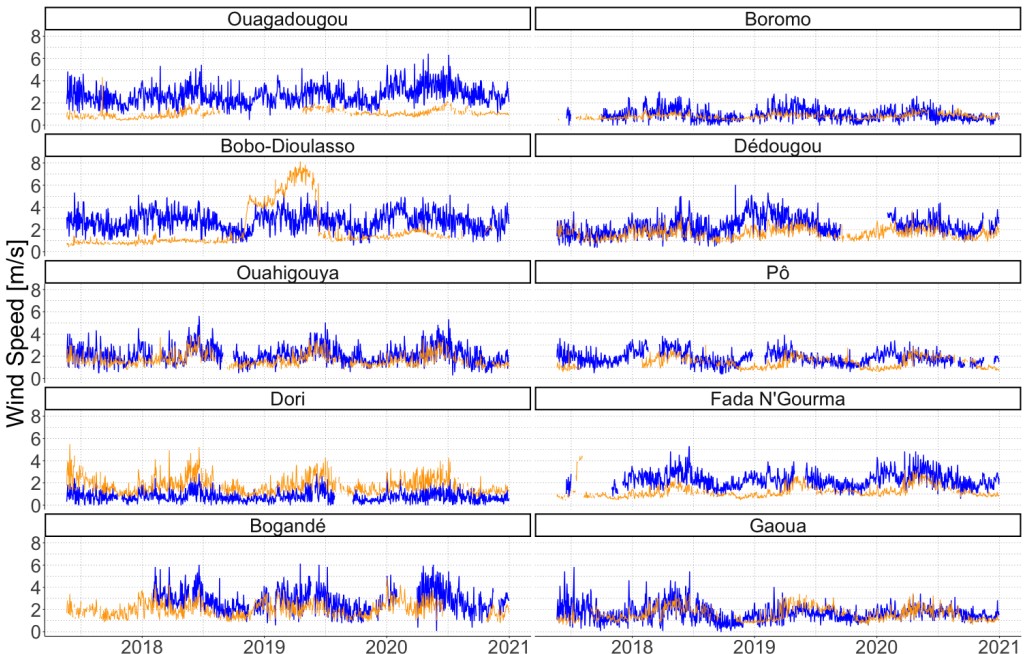

**Figure 16.** Comparison of wind speed time series between TAHMO stations in orange and the corresponding stations from meteorological service of Burkina Faso in blue.

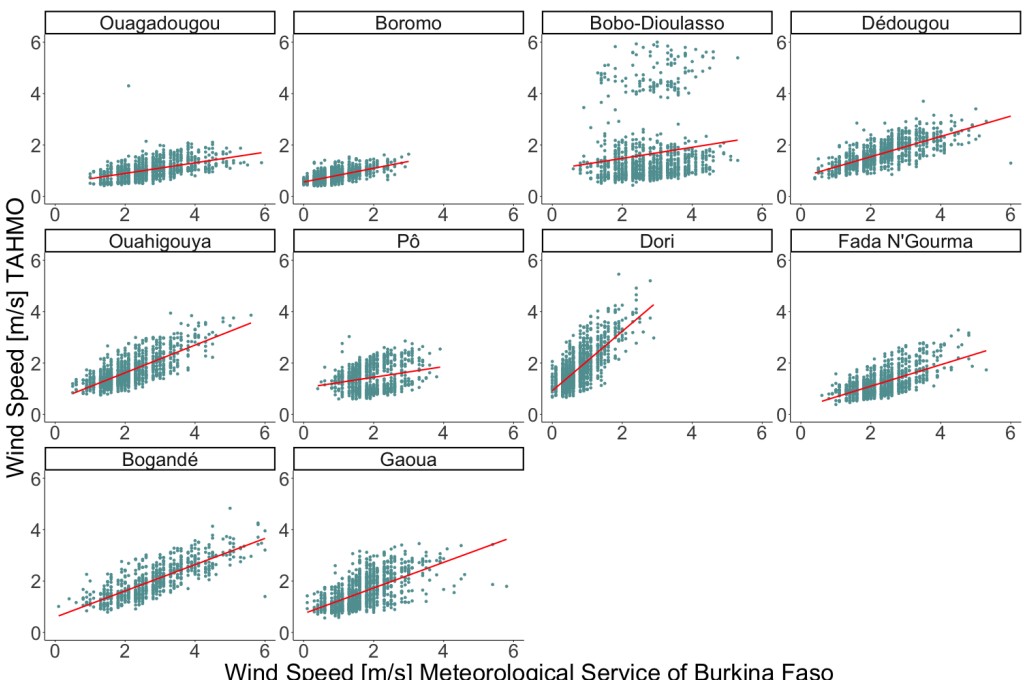

**Figure 17.** Scatterplot between wind speed time series of TAHMO stations (*y*-axis) and the corresponding stations from meteorological service of Burkina Faso (*x*-axis).

## 4. Summary and Conclusions

This study evaluates the hydro-meteorological TAHMO data set for potential applications in West Africa, with a spatial focus on Burkina Faso. Until now, to the best of the authors knowledge, TAHMO is applied for validation of satellite-derived data based on short series only, but has not itself undergone a documented validation while operating in African. Validation studies for TAHMO exist for Europe under different climatic conditions only [16,17]. It is important to stress that differences in the performances from these studies are also expected due to different weather regimes in Europe and Africa, such as the distinct rainy (and dry) seasons with higher rainfall intensities as well as comparatively higher temperature ranges in SSA. To fill this gap, the stations are examined for their availability, completeness, and plausibility, and finally for their quality and accuracy in comparison to meteorological reference stations. With respect to data availability, the number of MVs are below 15% on average for atmospheric pressure, temperature, precipitation, radiation, and relative humidity, and with MVs of up to 30% for wind measurements. For time series with low number of MVs, only small inconsistencies are found and could be discarded for statistical analyses based on initial plausibility checks. Considering the trade-offs between number of MVs and availability of reference stations in the near proximity, the performance analysis is finally focused on 10 stations which are relatively evenly distributed in BF. The performance analysis is done for temperature (minimum, maximum), precipitation, relative humidity, and wind speed. Temperature and relative humidity showed the most reliable results with high correlation coefficients. With respect to temperature, higher agreements and less scatter are observed for maximum temperature compared to minimum temperature. However, both time series reveal continuously slightly lower observed values compared to the reference station. For a longer period within the relative humidity series for Ouagadougou, a tendency towards non-plausible saturated values at 100% is observed. Moreover, TAHMO is slightly overestimating relative humidity, which in turn leads to an increase in evaporative cooling in the surrounding of the ATMOS 41 sensors, and thus leading to a correspondingly small and systematically reduced air temperatures. The high correlation coefficient is indicating that these biases are linear. Although the correlation coefficient for precipitation is lower than for temperature and relative humidity (on average around 0.7), the performance is considered reasonable in respect that the

spatial dependency is per se lower (compared to the variables with higher correlation coefficients). Both, the rainfall amounts as well as the spatio-temporal variability is well captured (spatial patterns due to the rainy and dry seasons, modulated by the movement of the ITCZ). For wind speed, the correspondence between TAHMO and the reference stations is lowest, which cannot be attributed alone to the inherently higher variability of wind. More complex bias structures (generally lower variability, lower magnitude of short-term fluctuations, non-consistent under- and overestimations) in TAHMO are leading to the observed low correlation coefficients and hamper the correction.

It is concluded that temperature and relative humidity obtained from TAHMO can potentially be applied in hydro-meteorological applications. Considering the performance of precipitation (correlation as well as accumulated sums), TAHMO data seem to be ready for hydro-meteorological applications. The use of wind speed, however, cannot be recommended, at least not without application of bias correction methods. Thus, with minor restrictions, TAHMO offers a reliable and cost-efficient solution for practical applications in hydro-meteorology. Currently, its usage is still restricted for climate (impact) research due to the limited length of the time series, but in the long run its continuously increasing spatial coverage will be a great asset. Another important asset is that the data set can be retrieved for entire SSA, and is not bound to the data distribution restrictions of each country.

This study is limited and could possibly be extended in the following aspects:

- the selection of the study region (BF) is based on the availability of suitable reference stations and MV statistics of the time series. Further studies for different regions across SSA (e.g., in East Africa) are needed to verify the conclusions of this study.
- the performance evaluation is restricted to the daily aggregation level. Subdaily analyses, e.g., based on hourly values or diurnal ranges, could provide valuable additional information for hydro-meteorological applications.
- under given restrictions in the availability of reference data, not all of the variables could be validated for TAHMO. Solar radiation as well as soil moisture are very crucial variables for hydrological and agricultural impact studies. Thus, a performance analyses of these variables of TAHMO data would be of great importance for ongoing and future research activities in SSA.
- it could not be quantified to which extent the differences between TAHMO stations and reference stations can be attributed to differences in the sensors used. For this purpose, systematic comparisons (located in immediate vicinity) between the different sensors are necessary under prevailing climate conditions in SSA.
- bias correction approaches, usually known from impact modelling based on climate model output (e.g., [27]) could potentially be applied to correct for the biases observed in some of the TAHMO variables (e.g., wind speed). Further studies analysing the skill of bias correction for TAHMO variables are needed.

**Author Contributions:** Analyses and figures, J.S.; conceptualization and first draft writing, P.L. and J.S.; review and editing, J.B., M.W., W.S., H.K.; funding acquisition, H.K. All authors have read and agreed to the published version of the manuscript.

**Funding:** This work was partly funded by the BMBF research projects EnerShelF (grant number 03SF0567D), SALDi (grant number 01LL1701B) and CONCERT (grant number 01LG2089A) of the WASCAL-phase-2 programme.

**Institutional Review Board Statement:** Not applicable.

**Informed Consent Statement:** Not applicable.

**Data Availability Statement:** The Trans-African Hydro-Meteorological Observatory (TAHMO) data can be retrieved from https://tahmo.org/ (accessed on 29 October 2021). Interested parties may contact info@tahmo.org for these data. The meteorological data used as reference in this study can be obtained from the National Meteorological Agency of Burkina Faso (ANAM-BF).

**Acknowledgments:** We thank the Trans-African Hydro-Meteorological Observatory (TAHMO) and the the National Meteorological Agency of Burkina Faso (ANAM-BF) for the provision of meteorological data. We acknowledge support by the KIT-Publication Fund of the Karlsruhe Institute of Technology.

**Conflicts of Interest:** The authors declare no conflict of interest.

## Appendix A. Additional Figures

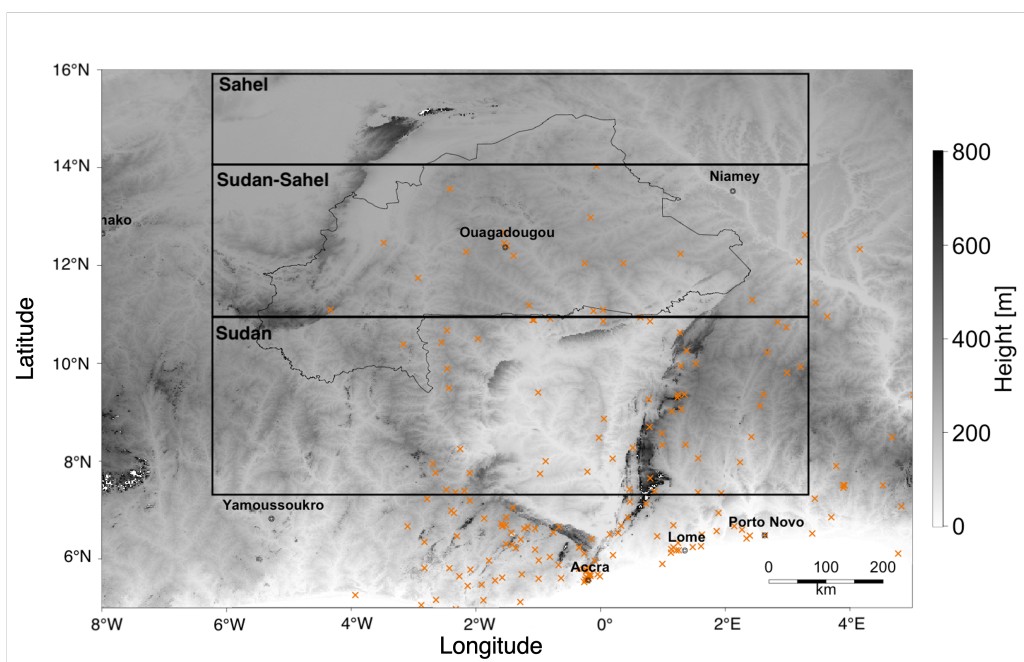

**Figure A1.** Location of the available TAHMO stations in West Africa.

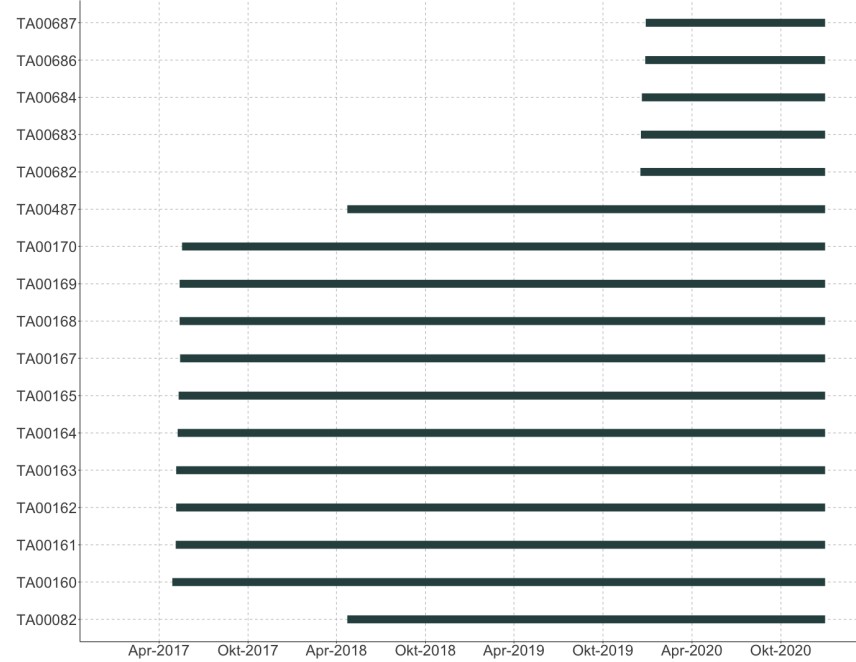

**Figure A2.** Begin of measurement period of the 17 preselected TAHMO stations in Burkina Faso, from which stations TA00160–TA00170 (starting from May 2017) are selected for further statistical analyses.

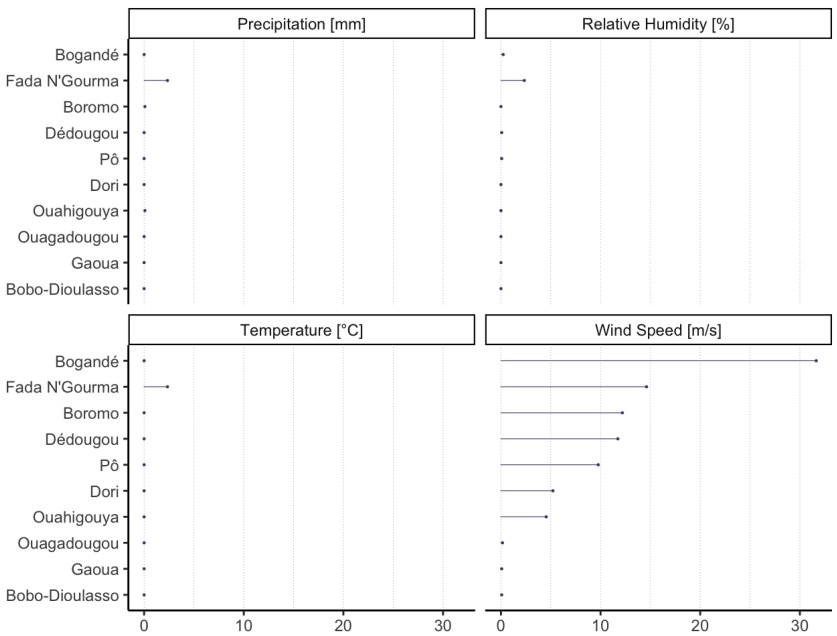

**Figure A3.** Number of missing values (MVs) [%] for all variables measured by the MET stations in Burkina Faso within the period 06/20/2017–12/31/2020.

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
