# Peer review of "Exploring the Potential of the Cost-Efficient TAHMO Observation Data for Hydro-Meteorological Applications in Sub-Saharan Africa"

_water, doi:10.3390/w13223308_

Round 1

Reviewer 1 Report

The authors present interesting and relevant work on the potential of TAHMO stations. The statistical analysis is straightforward and clearly presented yet would benefit from a more in-depth discussion. 

Suggestions:

  • discuss the effect of seasonality in the timeseries on the significance of the found correlations, e.g. by using climatology as a zero hypothesis 
  • if appropriate, discuss to which extent differences can be attributed to differences in sensors.
  • relate the results back to earlier results of TAHMO validation in Europe.

More detailed comments and suggestions are attached.

I would be happy to review an improved version of the manuscript.

Reviewer 2 Report

In section 2.2.1, Please discuss the main reasons behind missing values in these stations. Is there an ongoing problem? Any solution?

Add legend for Figures, especially for the given time series.

Has the same kind of precipitation gauges been used in MET and TAHOMA stations? Does the answer affect the results?
